# Synergistic Effect of Methyl Jasmonate and Abscisic Acid Co-Treatment on Avenanthramide Production in Germinating Oats

**DOI:** 10.3390/ijms22094779

**Published:** 2021-04-30

**Authors:** Soyoung Kim, Tae Hee Kim, Yu Jeong Jeong, Su Hyun Park, Sung Chul Park, Jiyoung Lee, Kwang Yeol Yang, Jae Cheol Jeong, Cha Young Kim

**Affiliations:** 1Biological Resource Center, Korea Research Institute of Bioscience and Biotechnology (KRIBB), Jeongeup 56212, Korea; cherish767@kribb.re.kr (S.K.); xogml5344@naver.com (T.H.K.); yjjeong@kribb.re.kr (Y.J.J.); suhyun@kribb.re.kr (S.H.P.); heypsc@kribb.re.kr (S.C.P.); jiyoung1@kribb.re.kr (J.L.); 2Department of Plant Biotechnology, College of Agriculture and Life Science, Chonnam National University, Gwangju 61186, Korea; 3Department of Applied Biology, College of Agriculture and Life Science, Chonnam National University, Gwangju 61186, Korea; kyyang@jnu.ac.kr

**Keywords:** ABA, *Avena sativa*, elicitation, MeJA, metabolite farming

## Abstract

The oat (*Avena sativa* L.) is a grain of the Poaceae grass family and contains many powerful anti-oxidants, including avenanthramides as phenolic alkaloids with anti-inflammatory, anti-oxidant, anti-itch, anti-irritant, and anti-atherogenic activities. Here, the treatment of germinating oats with methyl jasmonate (MeJA) or abscisic acid (ABA) resulted in 2.5-fold (582.9 mg/kg FW) and 2.8-fold (642.9 mg/kg FW) increase in avenanthramide content, respectively, relative to untreated controls (232.6 mg/kg FW). Moreover, MeJA and ABA co-treatment synergistically increased avenanthramide production in germinating oats to 1505 mg/kg FW. Individual or combined MeJA and ABA treatment increased the expression of genes encoding key catalytic enzymes in the avenanthramide-biosynthesis pathway, including hydroxycinnamoyl-CoA:hydrocyanthranilate N-hydroxycinnamoyl transferase (HHT). Further analyses showed that six AsHHT genes were effectively upregulated by MeJA or ABA treatment, especially AsHHT4 for MeJA and AsHHT5 for ABA, thereby enhancing the production of all three avenanthramides in germinating oats. Specifically, AsHHT5 exhibited the highest expression following MeJA and ABA co-treatment, indicating that AsHHT5 played a more crucial role in avenanthramide biosynthesis in response to MeJA and ABA co-treatment of germinating oats. These findings suggest that elicitor-mediated metabolite farming using MeJA and ABA could be a valuable method for avenanthramide production in germinating oats.

## 1. Introduction

Plants are complex organisms that form an important part of our daily lives. In addition to primary metabolites essential for growth and development, plants produce a vast number of compounds that play crucial roles in defense and environmental adaptation (i.e., secondary metabolites) [1,2,3,4]. Secondary metabolites, such as terpenes, alkaloids, flavonoids, and carotenoids, have many industrial uses, including as pharmaceuticals, pesticides, flavorings, and food additives [2,3]. Oats produce a group of phenolic secondary metabolites known as avenanthramides, which are low-molecular-weight polyphenols produced only in oats [5] and act as phytoalexins produced in oat leaves in response to pathogen infection [6] or treatment with elicitors [7,8,9].

Avenanthramides are endowed with beneficial health properties and possess antioxidant, antipruritic, anti-proliferative, anticancer, and anti-inflammatory effects [10,11,12,13,14,15]. Moreover, avenanthramides exhibit strong antioxidant properties with up to 30-fold greater activity than other phenolic compounds produced in oats [16,17,18]. To date, more than 25 different types of avenanthramides have been identified in oats [19,20,21]. Avenanthramides are a group of alkaloids that comprise an anthranilic acid derivative linked to a hydroxycinnamic acid derivative by an amide bond [5,19,22]. Three major avenanthramide isoforms [avenanthramide (Avn) A, B, and C] have been identified so far [10], all of which contain 5-hydroxyanthranilic acid and various hydroxycinnamic acids (p-coumaric acid in Avn A, ferulic acid in Avn B, and caffeic acid in Avn C) [19,23,24]. Avenanthramide biosynthesis follows the phenylpropanoid pathway (Figure 1), with L-phenylalanine representing the starting amino acid for biosynthesis [25]. In the first step, phenylalanine is transformed into cinnamic acid by phenylalanine ammonia-lyase (PAL) before being converted into 4-coumaric acid, caffeic acid, and ferulic acid by cinnamate 4-hydroxylase enzymes [25]. In the next step, 4-coumarate-CoA ligase (4CL) catalyzes the condensation of 4-coumaroyl-CoA and two malonyl-CoA molecules. The phenylpropanoid pathway provides precursors for many secondary metabolites in plants, including lignins, anthocyanins, and flavonoids [26]. Avenanthramides are synthesized through a condensation process involving hydroxyanthranilic acid and hydroxycinnamoyl-CoA, which is catalyzed by the key enzyme hydroxycinnamoyl-CoA:hydroxyanthranilate N-hydroxycinnamoyl transferase (HHT) [7,8,21]. To date, six *HHT* and one caffeoyl-CoA *O*-methyltransferase (*CCoAOMT*) cDNAs have been isolated from oats [8]. The encoded isoforms of AsHHT1–3 (~95 to 98%) and AsHHT4–6 (~95%) share high amino acid sequence identity, whereas isoforms of AsHHT4–6 exhibit somewhat low amino acid sequence identity (~82%) with AsHTT1–3 [21]. A previous study reported that expression of *AsHHT1* and *CCoAOMT* is concomitantly induced along with phytoalexin accumulation by a vitorin elicitor in *Pc-2*/*Vb* oat lines [8].

Metabolite farming is an agricultural method or technology used to induce the production of secondary metabolites through the use of various elicitors in plants, cells, tissues, and organs [3,27,28,29,30]. The aim is to increase the value of plants or plant cell cultures by enhancing the contents of the main bioactive substances. These methods are applied to generate useful natural materials in high-value-added foods, cosmetics, and pharmaceutical products and as a key approach for enhancing the competitiveness of agriculture as an industry [31].

In this study, the application of a small-scale metabolite-farming system to increase the avenanthramide content in germinating oats via treatment with various elicitors was investigated. The results suggested that co-treatment of germinating oats with methyl jasmonate (MeJA) or abscisic acid (ABA) showed a significant and synergistic effect on the expression of structural genes involved in avenanthramide biosynthesis, thereby leading to enhanced production of avenanthramides. This work would provide a basis for applications of elicitor-mediated metabolite farming to develop high-value-added plants capable of increasing the accumulation of bioactive metabolites.

## 2. Results

### 2.1. Avenanthramide Content of Germinating Oats

In the present study, avenanthramide contents in a time-course experiment over 5 days after sowing were analyzed. Figure 2A shows the germinating oat phenotype from days 0 to 5 of culture. Radicles emerged 1 day after sowing, and the sprouts grew to lengths of up to 5 cm in 5 days. The contents of Avn A and Avn C increased 9.8- and 6.9-fold, respectively, during the day 1, and the total avenanthramide content increased 5.6-fold (205.7 mg/kg FW) relative to that on day 0 (36.5 mg/kg FW). On day 2, the total avenanthramide content had increased 8.5-fold relative to day 0. Moreover, the highest proportions of Avn A (157.3 mg/kg FW), Avn B (154.2 mg/kg FW), and Avn C (152.4 mg/kg FW) were quantified on day 3 (Figure 2B). Therefore, day 2 was selected as the optimal time point for elicitor treatment to enhance avenanthramide content during oat germination.

### 2.2. Induction of Avenanthramide Production and Expression of Avenanthramide-Biosynthetic Genes in Germinating Oats Following Treatment with Various Elicitors

Two-day-old germinating oats were treated for 72 h with various elicitors, such as methyl viologen (MV; 10 μM), MeJA (100 μM), ABA (300 μM), chitosan (Chi; 200 μg/mL), salicylic acid (SA; 100 μM), and ethephon (ET; 300 μM), and 5-day-old germinating oats were collected and analyzed by high-performance liquid chromatography (HPLC) (Figure 3A). MeJA and ABA effectively enhanced the production of avenanthramides in germinating oats, with MeJA treatment resulting in increases of ~2.7-fold (190.1 mg/kg FW) for Avn A, 2.5-fold (260.4 mg/kg FW) for Avn B, 2.3-fold (132.4 mg/kg FW) for Avn C, and 2.5-fold (582.9 mg/kg FW) for total avenanthramides relative to the ethanol (EtOH)-treated control (Avn A, 71.1 mg/kg FW; Avn B, 103.2 mg/kg FW; Avn C, 58.3 mg/kg FW; and total avenanthramides, 232.6 mg/kg FW). Additionally, ABA-treated seedlings showed an increase in avenanthramide production of 2.7-fold (193.8 mg/kg FW) for Avn A, 2.0-fold (207.4 mg/kg FW) for Avn B, 4.1-fold (241.7 mg/kg FW) for Avn C, and 2.8-fold (642.9 mg/kg FW) for total avenanthramides relative to the EtOH control. No other elicitors induced significantly elevated levels of avenanthramide production relative to the EtOH and distilled water (DW) controls. These results suggested the efficacy of MeJA and ABA as elicitors for enhanced production of avenanthramides in oat seedlings.

The expression levels of genes encoding enzymes (PAL, 4CL, CCoAOMT, and HHT) involved in the avenanthramide-biosynthesis pathway (Figure 1) were determined by quantitative reverse transcription (qRT)-PCR, with the AsActin gene used as an internal control. The expression of *AsPAL, As4CL*, and *AsCCoAOMT* in germinating oats was significantly induced by treatments of MeJA and ABA, with 4.2- and 19.7-fold increases in *AsPAL*, 3.0- and 2.8-fold increases in *As4CL*, and 2.3- and 1.9-fold increases in *AsCCoAOMT*, respectively, relative to the negative control (0.1% EtOH) (Figure 3B,C). The six *AsHHT* genes examined effectively responded to treatments of MeJA and ABA (1.7- to 10.8-fold increases in production). In particular, MeJA treatment increased *AsHHT4* expression 10.8-fold, and ABA treatment significantly upregulated *AsHHT5* expression 6.2-fold in germinating oats. Additionally, SA and ET treatments slightly upregulated *AsPAL*, *As4CL*, *AsCCoAOMT*, *AsHHT1*, *AsHHT2*, *AsHHT3*, and *AsHHT5* expression, whereas those treatments reduced *AsHHT4* and *AsHHT6* expression. However, MV and Chi treatments did not significantly affect the expression of avenanthramide-biosynthetic genes. These results indicated that MeJA and ABA treatments of germinating oats effectively enhanced the expression of the six *AsHHT* genes analyzed in this study.

### 2.3. Enhanced Production of Avnenathramides by MeJA and/or ABA Treatment

To determine the optimal concentration of MeJA for enhancing the total avenanthramide yield in germinating oats, 2-day-old seedlings were treated with various MeJA concentrations for 3 days. The accumulation of avenanthramides in the seedlings gradually increased along with the MeJA concentration, reaching a peak at 75 μM MeJA (578.61 mg/kg FW), which was a 2.3-fold increase relative to that in control oats and resulting in Avn A, B, and C contents of 179.8 mg/kg FW, 268.7 mg/kg FW, and 130.0 mg/kg FW, respectively (Figure 4B). To determine the optimal concentration of ABA, 2-day-old oats were treated with various ABA concentrations (range: 10–300 µM), and whole plants were collected on day 3 after treatment. Figure 4C shows that ABA treatment at all concentrations significantly increased the production of avenanthramides in the germinating oats relative to the control. In particular, Avn C production increased >4.0-fold at all concentrations, suggesting that ABA played an important role in Avn C accumulation. Moreover, compared with the control, the maximum level of produced avenanthramides was observed from germinating oats treated with 50 μM ABA (a 3.4-fold increase; 828.9 mg/kg FW).

### 2.4. Synergistic Effect of MeJA and ABA Co-Treatment on Avenanthramide Production in Germinating Oats

The results demonstrated that MeJA or ABA can induce the production of avenanthramides in geminating oats. Therefore, further experiments were conducted to determine whether co-treatment with MeJA and ABA might have a synergistic effect on avenanthramide production (Figure 5). Two-day-old germinating oats were treated for 3 days with various concentrations of MeJA and ABA based on the concentrations used in the experiments reported in Figure 3. Combined MeJA and ABA treatment increased the production of avenanthramides at all concentrations tested. Specifically, compared with the control, the highest combined yields of Avn A (515.6 mg/kg FW), Avn B (549.9 mg/kg FW), and Avn C (439.8 mg/kg FW) were observed in oats treated with 75 μM MeJA and 25 μM ABA (Figure 5B). These results confirmed that combined treatment with MeJA and ABA had a synergistic effect on increasing avenanthramide synthesis in germinating oats.

### 2.5. Expression of Avenanthramide-Biosynthesis-Related Genes in MeJA- and ABA-Co-Treated Oats

The expression of avenanthramide-biosynthesis-related genes was further compared after single treatments with MeJA (75 μM) or ABA (25 μM) and combined treatments with MeJA (75 μM) and ABA (75 μM). Oat samples treated with 0.1% EtOH were used as a negative control, and qRT-PCR was performed on 5-day-old germinating oats (Figure 6). qRT-PCR results showed synergistic upregulation of *AsPAL, As4CL*, and *AsHHT5* expression following co-treatment of germinating oats with MeJA and ABA, with *AsPAL* expression upregulated 2.5- and 2.8-fold by MeJA and ABA treatment, respectively, and 4.0-fold by co-treatment relative to the control. By contrast, *As4CL* expression increased only slightly by 1.5-fold in response to co-treatment. Furthermore, among the six *AsHHT*s genes analyzed in this study, *AsHHT5* exhibited the highest expression, with a 5.1-fold increase following co-treatment and 1.6- and 3.1-fold increases following separate MeJA and ABA treatments, respectively, relative to the control. However, the expression of *AsHHT1, AsHHT2, AsHHT3, AsHHT4*, and *AsHHT6* was reduced by separate MeJA or ABA treatment, and even co-treatment failed to increase their expression relative to the control. These results suggested that among the six *AsHHT*s genes, *AsHHT5* played a more crucial role in avenanthramide biosynthesis in response to MeJA and ABA treatment. Additionally, it was found that *AsHHT1* expression was relatively high as compared with that of the other *AsHHT* genes in the untreated germinating oats. In particular, *AsHHT3* and *AsHHT4* expression was undetectable in germinating oats, regardless of treatment status (with MeJA and ABA or untreated) (Appendix A). These findings suggested that the six *AsHHT* genes were effectively upregulated in response to high concentrations of MeJA (100 μM) and ABA (300 μM), whereas five *AsHHT* genes (except *AsHHT5*) failed to respond to relatively low concentrations of MeJA (75 μM) and ABA (25 μM).

### 2.6. Analysis of Avenanthramide Contents and Biosynthetic Gene Expression in Different Organs of Germinating Oats

The total contents of all three avenanthramides in different organs 3 days following co-treatment of 2-day-old germinating oats with MeJA and ABA as elicitors were further analyzed. Different organ samples (leaf, grain, and root) were collected from 5-day-old oat seedlings after co-treatment and were determined using HPLC. As shown in Table 1, a higher content of avenanthramides was detected in grains (420.7 mg/kg FW) relative to root (2.3 mg/kg FW) and leaf (13.2 mg/kg FW) organs. By contrast, total avenanthramide content was significantly increased by co-treatment with MeJA and ABA, with the avenanthramide contents in co-treated roots (15.6 mg/kg FW) and leaves (40.1 mg/kg FW) ~6.7- and ~3.0-fold higher than those in untreated roots and leaves, respectively. The highest content of all three avenanthramides (968.9 mg/kg FW) was observed in the co-treated grains, which was ~2.3-fold higher relative to that in untreated grains (420.7 mg/kg FW). Among the three avenanthramides examined in this study, Avn B was the most abundant (~46%) in co-treated grains (Avn A, ~32%; and Avn C, ~21%) (Table 1). Overall, these results showed that avenanthramide content was increased throughout the plant by combined treatment with MeJA and ABA, with the highest avenanthramide production observed in the grain.

To determine the relationship between avenanthramide production and its biosynthetic gene expression, the relative expression levels of avenanthramide-biosynthesis-related genes in untreated and co-treated oat tissues were compared (Appendix A). Among the three tissues, the highest expression of AsPAL, As4CL, AsHHT1, AsHHT2, AsHHT5, and AsHHT6 was observed in the grains, whereas AsCCoAOMT expression was higher in root tissues relative to the control. This was positively correlated with the high avenanthramide production observed in the grains (Table 1). Additionally, the expression of AsPAL, As4CL, AsCCoAOMT, and AsHHTs following co-treatment with MeJA (75 μM) and ABA (25 μM) relative to the untreated control was examined (Figure 7). The results showed that AsPAL was upregulated ~5.4-, ~2.0-, and ~10.0-fold in leaf, grain, and root tissues, respectively. Moreover, As4CL expression was upregulated ~1.6-fold in grains but reduced ~0.6-fold in roots following co-treatment, and AsCCoAOMT expression was decreased ~0.6-fold in co-treated roots relative to the control. Furthermore, AsHHT6 expression was slightly increased by ~1.5- and ~1.2-fold in leaves and roots, respectively, but reduced ~0.4-fold in grains following co-treatment and relative to the control. AsHHT5 expression was upregulated ~2.3-fold in grains but downregulated ~0.7-fold in roots relative to the control. However, AsHHT2, AsHHT3, and AsHHT4 expression was reduced following co-treatment in all the tissues examined, with AsHHT3 and AsHHT4 expression decreased by ~0.4- and ~0.3-fold, respectively, relative to the control. By contrast, AsHHT1 expression was slightly upregulated by ~1.3-fold in leaves but decreased ~0.4-fold in roots relative to the control. Among the six AsHHTs, the greatest increase in expression in response to co-treatment was observed in AsHHT5 in grains; therefore, this suggested that AsHHT5 might play a more important role in avenanthramide biosynthesis in germinating oats in response to MeJA and ABA co-treatment. Furthermore, this result was consistent with the highest contents of all three avenanthramides in co-treated grains (Table 1).

## 3. Discussion

Oats have gained considerable interest in the past decade due to their nutritional significance and health-beneficial effects, mainly as foodstuffs but also as pharmaceuticals [5,13,17,32]. Several studies report that avenanthramides have antioxidant properties and potential therapeutic benefits, including anti-inflammatory, antiproliferative, and antigenotoxic effects [11,13,15,17,19,33]. These health benefits make avenanthramides attractive for their potential use in cosmetic, nutraceutical, and therapeutic preparations. Therefore, it is important to enhance their levels in oats in order to provide consumers with raw materials of enhanced nutritional value. In this study, high-value-added oats capable of producing high levels of avenanthramides following the application of elicitor-mediated metabolite-farming technology were developed. The application of exogenous elicitors often causes an increase in the concentration of bioactive compounds in plants [3,27,28]. MeJA, ABA, SA, and ethylene are currently used as powerful elicitors to induce secondary metabolism in plants and plant cell cultures. For example, preharvest treatment of ethylene on soybean plants induces significant accumulation of dietary isoflavones (up to 13,854 mg/g DW) in the leaves [28]. Therefore, elicitor-mediated metabolite farming can be a valuable agricultural technology enabling increases in the amounts of bioactive metabolites generated in plants or food crops.

In this study, the highest content of total avenanthramides was detected on day 3 in germinating oats, with this content most abundant in grains and lowest in roots (Figure 2B and Table 1). This was why seedlings were used as the experimental stage for evaluating elicitor-mediated increases in avenanthramide content. Previous studies have reported that the majority of avenanthramides are present in the oat groat, with the highest content in the bran [34]. The biosynthesis of avenanthramides is known to increase during the germination process of oat grains [19,23,34]. Moreover, these compounds are shown to induce in response to pathogen infection and elicitor treatment [5,7,9]. Wise et al. [9] previously reported that avenanthramide levels in leaves and roots were enhanced by treatment of oat plants with benzothiadiazole (BTH) via root drench, with BTH application eliciting maximal levels (562.44 mg/kg FW) of total avenanthramide production in oat leaves.

Additionally, it was found that co-treatment with MeJA and ABA of germinating oats synergistically induced the highest levels of avenanthramide production (1505.3 mg/kg FW), which was substantially higher than those reported previously [35,36,37]. For example, Chu et al. [35] reported that the total content of avenanthramides generated in whole oat extracts ranged from 8.41 mg/kg to 150.81 mg/kg in varieties of oats from Canada (1.05–43.77 mg/kg for Avn A; 3.66–58.63 mg/kg for Avn B; and 3.7–48.41 mg/kg for Avn C). Tong et al. [36] reported that the total content of produced avenanthramides ranged from 4.4 mg/kg DW to 718.5 mg/kg DW in varieties of oats from China. Moreover, Li et al. [37] reported that the total concentration of avenanthramides produced in whole oat grains ranged between 22.1 mg/kg DW and 471.2 mg/kg DW, with the Avn B concentration (7.3–222.8 mg/kg DW) in oat grains higher than that of the other two avenanthramides (6.1–112.3 mg/kg DW for Avn A; and 6.2–136.2 mg/kg DW for Avn C). These results are in agreement with the ones in the present study, where Avn B comprised the largest fraction in both untreated and co-treated germinating oats (Table 1). By contrast, Wise et al. [9] reported that Avn A was the most highly produced avenanthramide of the four analyzed, with a maximum concentration of ~379 mg/kg FW in BTH-treated leaves. However, reports showed that the concentration and composition of avenanthramides are greatly influenced by genotype, growing environment, and growth conditions [24,34,35,36,37]. Furthermore, although BTH strongly upregulated avenanthramide production in oat plants [9], no significant effects of BTH application on oats in samples analyzed were identified in the present study (data not shown). These different responses to BTH might be the result of genotypic differences or different cultivars. Specifically, the naked oat cultivar was used in the present study, whereas the husked oat cultivar was used by Wise et al. [9]. Based on these findings, it can be speculated that BTH can induce avenanthramide biosynthesis in husked cultivars but not in naked ones. Therefore, future research is needed to describe the metabolic changes in different oat cultivars and reveal their underlying molecular causes. Moreover, additional investigations are necessary to determine whether avenanthramide accumulation is elevated in leaf tissues at different growth stages in response to BTH. Although BTH is an effective elicitor for increasing avenanthramide production in leaves of husked oat cultivars, because it is classified as a pesticide, its use is restricted, whereas the use eco-friendly elicitors for healthy food products is promoted. As a result, it is important to identify novel harmless elicitors for increasing the concentration of specific metabolites in food crops.

Several studies show that Avn C exerts greater antioxidant activity than Avn A and B [16,17,18,38,39]. Dhakal et al. [38] reported that Avn C from germinated oats exhibits anti-allergic and anti-inflammatory effects in mast cells, and Umugire et al. [39] showed that Avn C exerts a strong protective effect on hearing loss caused by noise trauma. These results suggest Avn C as a promising candidate as a therapeutic for treating allergic diseases and hearing damage, given that Avn C can be provided through dietary products derived from oats. To this end, the CRISPR/Cas9 gene-editing system can potentially be applied to create high-value-added oats with high Avn C content. Li et al. [21] recently demonstrated the molecular mechanisms associated with biosynthesis of three major avenanthramides in oats, suggesting that oat HHTs are only involved in the biosynthesis of Avn A and C but not Avn B, which is synthesized by oat CCoAOMT enzyme from *O*-methylation of Avn C. In the present study, it was found that the expression of *AsHHT* genes was induced by MeJA and ABA treatment, respectively, especially *AsHHT4* for MeJA and *AsHHT5* for ABA, thereby enhancing the production of all three avenanthramides in germinating oats. Furthermore, we found that the basal expression level of *AsCCoAOMT* was high (~0.1 to 0.2-fold expression relative to AsActin) in the leaves, grains, and roots of germinating oats regardless of elicitation (Appendix A), indicating constitutive gene expression. These findings suggest that oat *CCoAOMT* might be a good target for gene editing to increase Avn C content; however, the complete biosynthetic pathway of avenanthramides in oats remains to be elucidated.

In conclusion, this study provides a basis for elicitor-mediated metabolite farming to produce high-value-added oats capable of producing high levels of avenanthramides. The data suggested that germinating oats co-treated with MeJA and ABA were the most effective producers of all three avenanthramides (Avn A, B, and C) and demonstrated a strong correlation between enhancement of avenanthramide production and biosynthesis-gene activation (*AsPAL*, *As4CL*, and *AsHHT5*) in germinating oats. Therefore, this metabolite-farming method could also be applied to develop other high-value-added plants capable of generating high levels of bioactive compounds with possible applications in functional foods or pharmaceutical supplements.

## 4. Materials and Methods

### 4.1. Plant Materials and Growth Conditions

The naked oat cultivar Choyang was obtained from the National Institute of Crop Science, Development Administration in Korea. For the experiments, oat seeds were soaked in water for 12 h and then sterilized at 50 °C for 10 min in a water bath before being washed three times with autoclaved distilled water (DW) [40]. Sterilized oat seeds were sown in plant culture dishes (100 × 40 mm; SPL Life Sciences, Seoul, Korea) containing 7 mL of water and two sheets of filter paper and cold-stratified at 4 °C in the dark for 2 days. The seeds were incubated at 25 °C for germination with a 16 h/8 h light/dark cycle.

### 4.2. Treatment with Various Elicitors and Sample Collection

Two-day-old germinating oats (20 seedlings) were treated for 72 h with various elicitors, as previously described [29]. In brief, MeJA and ABA were dissolved in 96% (*v*/*v*) ethanol (EtOH), whereas MV, SA, ET, and Chi were dissolved in distilled water (DW). The elicitors of 10 μM MV, 100 μM MeJA, 300 μM ABA, 100 μM SA, 300 μM ET, and 200 μg/mL Chi were used. EtOH (0.1%) and DW were used as negative controls. Elicitor treatments were performed in plant culture dishes containing 7 mL of elicitor solution with two sheets of filter paper. The treated samples were harvested after 3 days of treatment, followed by immediate freezing in liquid nitrogen. The samples were then pulverized using a mortar and pestle for quantification of avenanthramides by HPLC and qRT-PCR analyses.

### 4.3. Quantification of Avenanthramides by HPLC

Avenanthramides were extracted by sonication (WUC-D10H; DAIHAN Scientific, Wonju, Korea) for 20 min from 0.1 g of sample (FW) in 1 mL of 80% (*v*/*v*) methanol at 50 °C. Following centrifugation at 11,000 rpm for 5 min, the supernatants were air-dried by nitrogen evaporation (N-EVAP 111; Organomation Associates, Inc., Berlin, MA, USA). The residue was dissolved in 200 μL of 80% methanol and filtered through a 0.2 μm PTFE membrane [34]. HPLC analysis was performed at 30 °C using an Agilent 1260 system equipped with a quaternary pump, a ZORBAX SB-C18 column (5 μM, 4.6 × 150 mm), and photodiode array detector (Agilent Technologies, Santa Clara, CA, USA). The mobile phase comprised buffer A (H2O with 5% acetonitrile and 0.1% formic acid) and buffer B (acetonitrile with 0.1% formic acid). A gradient in B of 13% to 30% over 30 min at a flow rate of 1 mL/min was employed [22] with an injection volume of 5 μL and a detection wavelength set to 340 nm. Standard avenanthramides purchased from Sigma-Aldrich (St. Louis, MI, USA) (#30366, Avn A; #93105, Avn B; and #36465, Avn C) were used for identification and quantification.

### 4.4. RNA Extraction and qRT-PCR Analysis

Harvested oat samples of whole seedlings or each organ (leaf, grain, and root) were finely ground using a mortar and pestle with liquid nitrogen. Total RNA was extracted via a modified version of the method described by Porebski et al. [41] using cetyltrimethylammonium bromide (Intron Biotechnology, Seongnam, Korea) and an RNA extraction kit (QIAGEN, Hilden, Germany). RNA concentration and purity were estimated by spectrophotometry (Nanodrop NS-11; DeNovix, Wilmington, DE, USA). qRT-PCR analysis was performed using an iQ SYBR Green supermix reagent (Bio-Rad Laboratories, Hercules, CA, USA) on a CFX96 real-time PCR system (Bio-Rad Laboratories). Specific primers (Appendix A) were designed, as described by Yang et al. [8]. The reactions were subjected to an initial denaturation step at 95 °C for 15 min, followed by 40 cycles of denaturation at 95 °C for 20 s and annealing at 60 °C for 40 s, with melting curve analysis performed within the same tube or plate. Transcript levels were calculated by normalization against *AsActin* (KP257585) as a reference control. Relative changes in gene-expression levels were determined by the 2^−ΔΔCt^ method [42].

### 4.5. Statistical Analysis

Data were analyzed using SPSS (v.22.0; IBM Corp., Armonk, NY, USA). One-way analysis of variance was performed and followed by Tukey’s multiple range post hoc test to compare data among experimental groups. Comparisons of the mean values between two groups were performed using Student’s *t*-tests. All values are expressed as the mean ± standard deviation of triplicate experiments, and a *p* value < 0.05 was considered statistically significant.

## Figures and Tables

**Figure 1 ijms-22-04779-f001:**
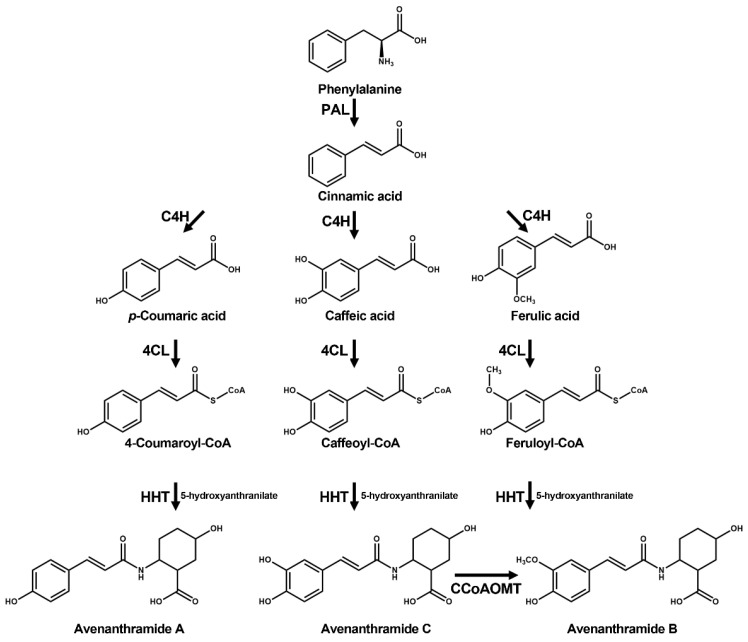
Proposed biosynthetic pathway of avenanthramides in oats. The avenanthramide-biosynthetic pathway was proposed based on information from the KEGG database (http://www.genome.jp/kegg/kegg2.html; accessed on 25 March 2020). PAL, phenylalanine ammonia lyase; C4H, cinnamate 4-hydroxylase; 4CL, 4-coumarate-CoA ligase; CCoAOMT, caffeoyl-CoA O-methyltransferase; HHT, hydroxycinnamoyl-CoA:hydroxyanthranilate N-hydroxycinnamoyl transferase; KEGG, Kyoto Encyclopedia of Genes and Genomes.

**Figure 2 ijms-22-04779-f002:**
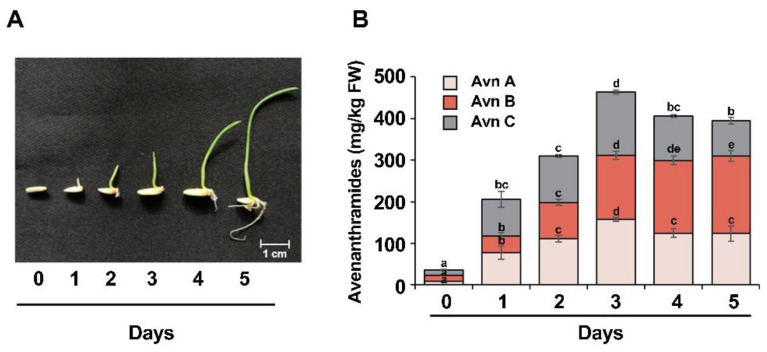
Yields of avenanthramides in oats at different stages of germination. (**A**) Oat seedlings at different developmental stages. Oat seeds were germinated in pre-wetted filter paper for 5 days. (**B**) Production of avenanthramides in oats during germination. Whole germinating oats were harvested on the days indicated after sowing. Twenty germinating oat plants were used for avenanthramide quantification at each time point, and the data shown are mean values. Data represent the mean of three independent replicates ± SD. Different letters above the bars indicates significant differences determined by ANOVA followed by a Tukey’s test (*p* < 0.05).

**Figure 3 ijms-22-04779-f003:**
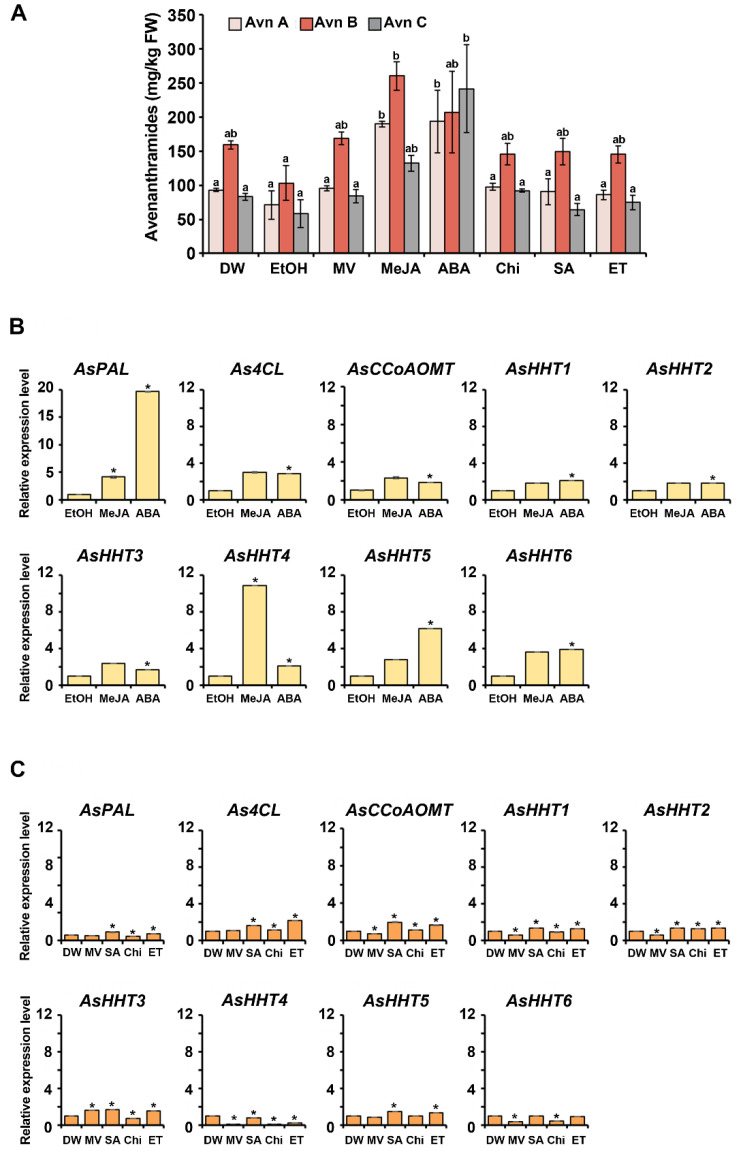
Avenanthramide content and relative expression levels of biosynthetic genes in germinating oats treated with various elicitors. Two-day-old oat seedlings were treated using various elicitors for 3 days. (**A**) Quantitative analysis of avenanthramide production by HPLC. Ethanol (0.1% EtOH) was used as a solvent control for MeJA and ABA, and distilled water (DW) was used as control for MV, Chi, SA, and ET. (**B**) The relative expression level of avenanthramide-biosynthetic genes under treatment with various elicitors dissolved in ethanol (0.1% EtOH). (**C**) The relative expression level of avenanthramide-biosynthetic genes under treatment with various elicitors dissolved in distilled water (DW). Data represent the mean of three independent replicates ± SD. Student’s *t*-test; * *p* < 0.05.

**Figure 4 ijms-22-04779-f004:**
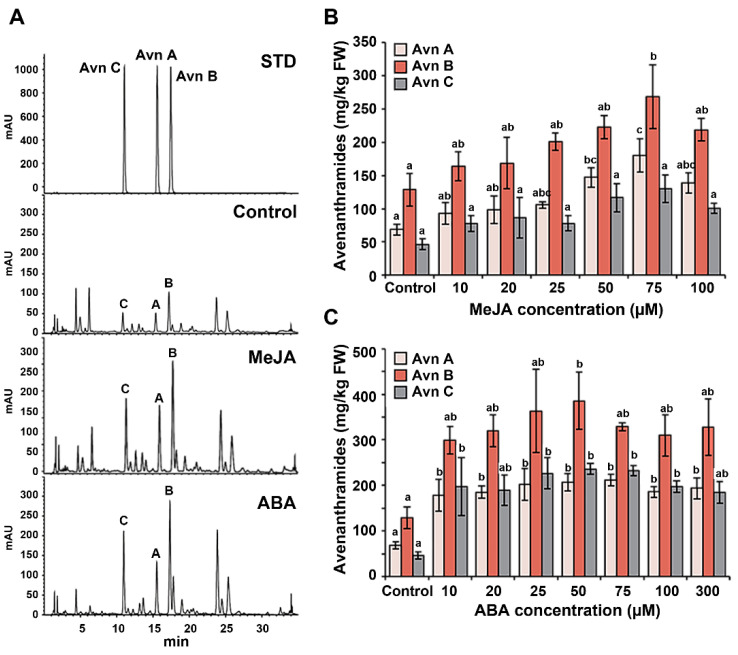
Effect of MeJA or ABA concentration on avenanthramide production in germinating oats. Two-day-old germinating oats were treated with different concentrations of MeJA or ABA for 3 days. (**A**) Results of HPLC analysis with UV detection at 340 nm. From top to bottom, chromatograms of STDs, control oats, and oats treated with MeJA (75 µM) and/or ABA (50 µM). The chromatogram of the STD samples shows retention times of 15.511 min, 17.320 min, and 10.964 min for the Avn A, Avn B, and Avn C STDs, respectively. (**B**,**C**) Quantitative measurement of avenanthramide levels in germinating oats treated with (**B**) MeJA or (**C**) ABA. Ethanol (0.1%) was used as the control. MeJA, methyl jasmonate; ABA, abscisic acid; Avn, avenanthramide; STD, standard. Different letters above the bars indicates significant differences determined by ANOVA followed by a Tukey’s test (*p* < 0.05).

**Figure 5 ijms-22-04779-f005:**
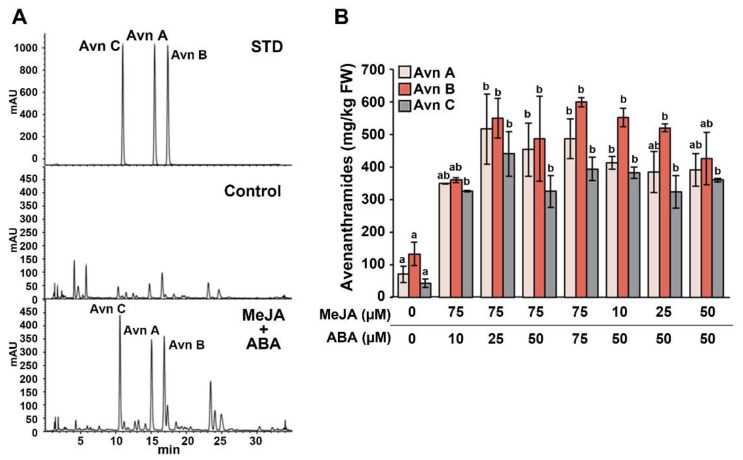
Enhanced production of avenanthramides by combined treatment with MeJA and ABA. Two-day-old germinating oats were treated with different concentrations of MeJA and ABA for 3 days. (**A**) Results of HPLC analysis with UV detection at 340 nm. Chromatograms of STDs (top), control oats (middle), and oats receiving combined treatment (75 µM MeJA + 25 µM ABA). The chromatogram of the STD samples shows retention times of 15.511 min, 17.320 min, and 10.964 min for the Avn A, Avn B, and Avn C STDs, respectively. (**B**) Avenanthramide levels in germinating oats receiving combined treatment with MeJA and ABA. Ethanol (0.1%) was used as the control. MeJA, methyl jasmonate; ABA, abscisic acid; Avn, avenanthramide; STD, standard. Different letters above the bars indicates significant differences determined by ANOVA followed by a Tukey’s test (*p* < 0.05).

**Figure 6 ijms-22-04779-f006:**
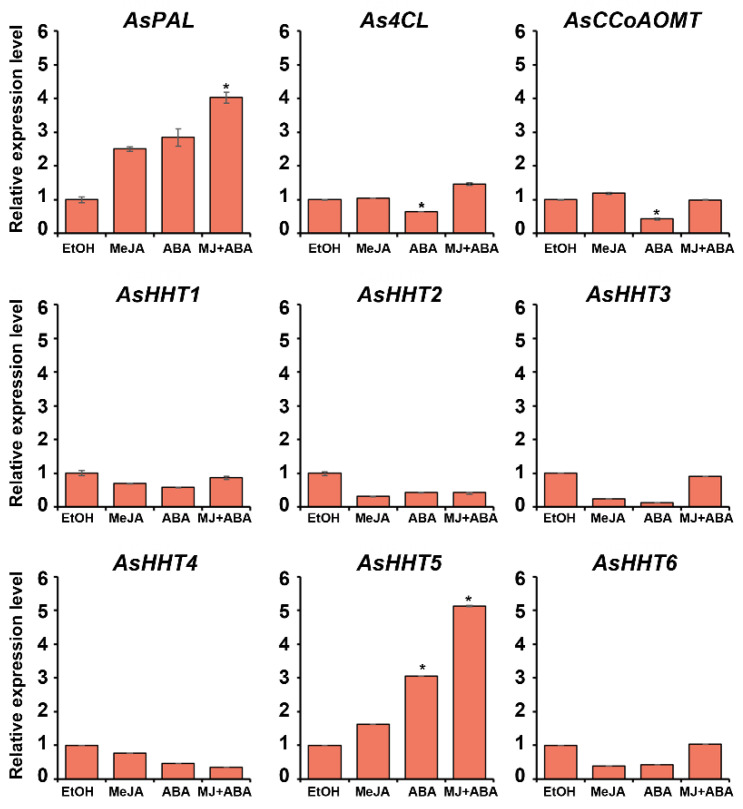
The relative expression of avenanthramide-biosynthetic genes following separate and combined treatment with MeJA and ABA of germinating oats. Transcript levels of *PAL, 4CL, CCoAOMT, HHT1, HHT2, HHT3, HHT4, HHT5*, and *HHT6* were analyzed by qRT-PCR in germinating oats treated with 75 μM MeJA and 25 μM ABA for 3 days. Relative expression levels were normalized to that of *AsActin* (KP257585) and are presented as fold induction relative to the control. Data represent the mean of three independent replicates ± SD. Student’s *t*-test; * *p* < 0.05. qRT-PCR, quantitative reverse transcription PCR; SD, standard deviation; MJ, methyl jasmonate.

**Figure 7 ijms-22-04779-f007:**
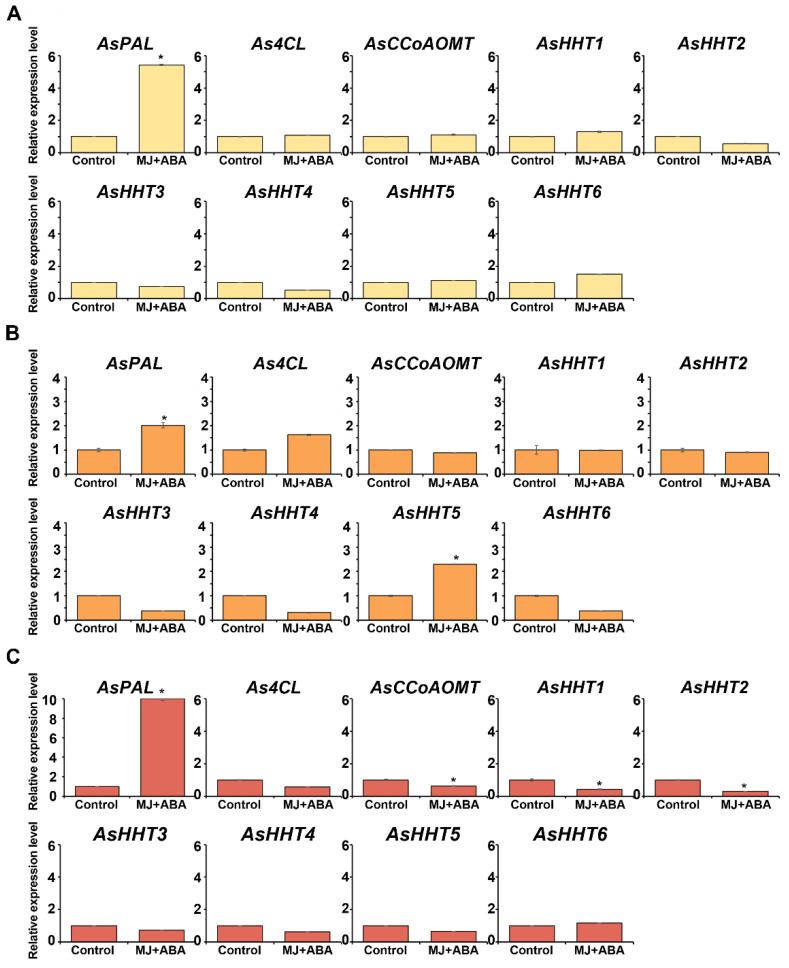
The relative expression levels of avenanthramide-biosynthetic genes in three types of oat organs: (**A**) leaves, (**B**) grains, and (**C**) roots. Combined treatment (MeJA + ABA) comprised 75 μM MeJA and 25 μM ABA. Two-day-old germinating oats were treated for 3 days and different organs were collected as leaves, grains, and roots. Transcript levels of *PAL*, *4CL*, *CCoAOMT*, *HHT1*, *HHT2*, *HHT3*, *HHT4, HHT5*, and *HHT6* were analyzed by qRT-PCR. Relative expression levels were normalized against AsActin (KP257585) and are presented as fold induction relative to the control. Data represent the mean of three independent replicates ± SD. Student’s *t*-test; ** p* < 0.05. qRT-PCR, quantitative reverse transcription PCR; SD, standard deviation; MJ, methyl jasmonate.

**Table 1 ijms-22-04779-t001:** Analysis of avenanthramide contents in different organs of germinating oats.

	Organs	Avn A (mg/kg FW)	Avn B (mg/kg FW)	Avn C (mg/kg FW)	Total Avenanthramides (mg/kg FW)
**Untreated** **control**	**Leaves**	2.9 ± 1.7 ^a^	6.2 ± 0.4 ^a^	4.1 ± 2.1 ^a^	13.2 ± 3.9 ^a^
	**Grains**	112.4 ± 22.7 ^b^	235.9 ± 41.5 ^b^	72.5 ± 23.8 ^a^	420.7 ± 85.1 ^b^
	**Roots**	0 ± 0 ^a^	2.3 ± 2.3 ^a^	0 ± 0 ^a^	2.3 ± 2.3 ^a^
**MeJA + ABA ***	**Leaves**	9.2 ± 2.9 ^ab^	21.8 ± 8.5 ^a^	9.1 ± 0.2 ^a^	40.1 ± 11.5 ^a^
	**Grains**	312.7 ± 51.1 ^c^	449.1 ± 78.6 ^c^	207.1 ± 31.7 ^b^	968.9 ± 160.7 ^c^
	**Roots**	3.9 ± 1.9 ^ab^	10.1 ± 5.1 ^a^	1.6 ± 1.6 ^a^	15.6 ± 8.1 ^a^

* Combined treatment with 75 μM MeJA and 25 μM ABA. Statistical differences among experimental conditions are labeled with different letters (*p* < 0.05).

## Data Availability

The data presented in this study are available on request from the corresponding author. Appendix A can be found at https://zenodo.org/record/4682677 (accessed on 13 April 2021).

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
