# Peer review of "Synergistic Effect of Methyl Jasmonate and Abscisic Acid Co-Treatment on Avenanthramide Production in Germinating Oats"

_ijms, 2021, doi:10.3390/ijms22094779_

Round 1

Reviewer 1 Report

This work reports the elicitor effects of several compounds in the production of Avenanthramide by oat grins and seedlings.

The manuscript is interesting por publication but there are several items that must be addressed before the ms could be accepted for publication.

I've made the comments and corrections directly on the manuscript.

Author Response

Reviewer 1:

Comments and suggestions for authors:

This work reports the elicitor effects of several compounds in the production of avenanthramide by oat grains and seedlings. The manuscript is interesting for publication but there are several items that must be addressed before the ms could be accepted for publication.

I've made the comments and corrections directly on the manuscript.

â–¶ Response:

As the reviewer’s comment, we have revised all the comments and suggestions given by the reviewer throughout the whole manuscript. Please check the revised manuscript.

Reviewer 2 Report

The manuscript by Kim et al. focuses on the study of the effect of different elicitors in oat germinates' content of avenathramide. Authors found that MeJa and ABA had a particular effect in avenathamide production and studied their independent and also their synergistic effect. The study  is well structured and the manuscript is clearly written.

However some figures lack statistical analysis, which hinder the results interpretation and which treatments are the controls of the experiments is not always well explained. Furthermore, it is urgent that the authors add more details in terms of the number of biological replicates used in the study in the M&M. 

Keywords shouldn't repeat words from the title; please change.

Replace the last paragraph of the introduction section to present your hypothesis instead of the obtained results. Focus the hypothesis on the target utilization of your findings.

Figure 2B, 3 and 6 lack statistical analysis.

Figure 3A legend must include the meaning of each treatment presented in the XX axis; which treatments are the controls must be clearly shown.

There are several sections in the Results that are discussion of the results. Authors should remove all text discussing results and correlating figures, as this should be presented seperately in the Discussion. In this way, the results will be more focused and simpler and repetition between sections will be avoided. 

In 4.2 section the number of plants per treatment must be specified as well as the number that were analysed by HPLC or by qPCR.

Please note the additional remarks in the pdf file.

Round 2

Reviewer 2 Report

Thank you for incorporating all suggestions.

Just one minor comment: In Figure 3, although the authors have added some information to the legend, it still lacks the definition of each abbreviation. A Figure must be self-explanatory and if one doesn't read the text it is impossible to understand the abbreviations.